# Unusual Observations in Leishmaniasis—An Overview

**DOI:** 10.3390/pathogens12020297

**Published:** 2023-02-10

**Authors:** Priya Yadav, Mudsser Azam, V Ramesh, Ruchi Singh

**Affiliations:** 1ICMR-National Institute of Pathology, New Delhi 110029, India; 2Manipal Academy of Higher Education, Manipal 576104, India; 3Department of Dermatology, ESIC Hospital, Faridabad 1210026, India

**Keywords:** leishmaniasis, visceral leishmaniasis, cutaneous leishmaniasis, mucocutaneous leishmaniasis, post-kala-azar dermal leishmaniasis, atypical leishmaniasis, rare presentations of leishmaniasis, immunocompromised

## Abstract

Leishmaniasis significantly affects the population of the tropics and subtropics. Clinical features and infective species of *Leishmania* are the primary factors driving the direction of diagnosis. The rise in incidences of atypical presentations present a challenge in patient treatment. Knowledge of unusual/rare presentations can aid in having a broader perspective for including the different aspects during the examination and thus avoid misdiagnosis. A comprehensive literature survey was performed to present the array of atypical presentations confounding clinicians which have been seen in leishmaniasis. Case reports of unusual findings based on the localizations and morphology of lesions and infective species and the predominant geographical sites over almost five decades highlight such presentations in the population. Information regarding the clinical features recorded in the patient and the chosen treatment was extracted to put forward the preferred drug regimen in such cases. This comprehensive review presents various unusual observations seen in visceral leishmaniasis, post-kala-azar dermal leishmaniasis, cutaneous leishmaniasis, and mucocutaneous leishmaniasis. It highlights the need to consider such features in association with differential diagnosis to facilitate proper treatment of the patient.

## 1. Introduction

The World Health Organization (WHO) categorized leishmaniasis as a neglected tropical disease as it affects public health and is concealed negligence in the development of an adequate and affordable cure. With millions of cases every year across the planet, leishmaniasis is still a rampant disease. Special emphasis was given to the measures on filling in the gaps in the diagnosis and management of leishmaniasis in the 14th meeting of the WHO’s Strategic and Technical Advisory Group for Neglected Tropical Diseases [1]. The transmission of leishmaniasis either occurs through animals (zoonotic) or humans (anthroponotic). Although transmission through blood transfusions, placental transference, and shared needles is rare, a few incidences have been reported [2,3,4]. Visceral leishmaniasis (VL), cutaneous leishmaniasis (CL), and mucocutaneous leishmaniasis (MCL) are the clinical forms. Approximately 0.6–1.0 million new cases of CL and 0.5–0.9 million cases of VL occur globally on an annual basis [5]. In regions endemic for CL, the progression to MCL is approximately 20% [6]. Countries such as Bolivia, Brazil, Peru and Ethiopia present more than 90% of MCL cases. Post-kala-azar dermal leishmaniasis (PKDL) is a dermal sequela of VL developing in 5–15% of treated cases. More than 70 species of animals (including humans) have been classified as host reservoirs of leishmaniasis [5].

*Leishmania* is classified into four subgenera—*Leishmania*, *Sauroleishmania* (reptile-infecting), *Mundinia* and *Viannia*. Subgenus *Mundinia* has recently been described in the taxonomical hierarchy of *Leishmania*. This subgenus contains two taxa, namely *L. (Mundinia) orientalis* and *L. (Mundinia) martiniquensis*. The taxon supersedes the taxonomically invalid *L. siamensis* [7]. Each species bears its own spatial and host preferences and characteristic symptoms during infection [8]. *Leishmania donovani* complex (*L. donovani, L. infantum*), *L. major*, *L. tropica* and *L. aethiopica* are the Old World (OW) species, transmitted by *Phlebotomus* spp. *L. mexicana* complex (*L. mexicana, L. venezuelensis, and L. amazonensis*), whilst the species of the *L. viannia* subgenus (*L. braziliensis, L. guyanensis, L. panamensis, L. peruviana*) complex constitute the New World (NW) species, transmitted by *Lutzomyia* species [9,10]. The expansion of competent vector species affects the geographical distribution of parasites. Low socio-economic development, poverty, and lack of hygiene and amenities such as proper housing and residence in closer association with vegetation complement a more remarkable presentation of diseases such as VL [11].

### Leishmaniasis: Typical Manifestations and Leishmania spp.

*Leishmania* spp. infects the host and the immune response generated in response to the infection determines the type of disease manifestation. The macrophage internalizes the promastigotes of *Leishmania* through a range of macrophage receptors. Through these receptors, different species of *Leishmania* aim at various downstream signalling pathways and modulate the course of infection. Alteration in the milieu of chemokines and chemokine receptors activated through immune effector cell recruitment is another responsible factor in the progression of disease severity [12]. Variations within the species, at both chromosome and gene levels, influence the disease outcome and treatment response in patients [13]. Quantitative trait loci in the host gene affect the susceptibility to developing leishmaniasis [14].

VL is primarily caused by OW species, namely *L. donovani* (anthroponotic) and *L. infantum* (zoonotic). PKDL is a dermal sequela of VL appearing as a “macular, papular or nodular rash usually on the face, upper arms, trunks and other parts of the body”. In parts of Thailand and Myanmar, *L. (Mundinia) martiniquensis* infections are the primariy cause of VL [15]. *L. major* and *L. tropica* are primarily responsible for the localized form of CL (LCL) in OW regions, whereas *L. braziliensis* and *L. mexicana* for the localized form of CL (LCL) in NW regions. Zoonotic (wet/rural/early ulcerative) CL is caused by *L. major* and the anthroponotic infection (dry/urban/late ulcerative) is due to *L. tropica*. CL causes “skin lesions, mainly ulcers, on exposed parts of the body, leaving life-long scars and serious disability or stigma”. *L. braziliensis, L. amazonensis* and *L. panamensis* are species generally responsible for MCL that leads to the “partial or total destruction of the mucous membranes of the nose, mouth, and throat” due to the infiltration of amastigotes from the dermis to the mucosa layer of the oro-naso-laryngeal and tracheal regions [5]. Table 1 represents the common and rare species responsible for causing leishmaniasis [16,17].

The interspecies and intraspecies genetic exchange has been experimentally documented in the nature and may contribute to new phenotypic traits. Apart from contributing to the genomic diversity of the parasite, interspecies hybrids of *Leishmania* such as those formed between *L. donovani* and *L. major*/*L. tropica* lead to the emergence of novel genotypes of Leishmania, consequently affecting the transmission of CL in countries such as Sri Lanka [18].

## 2. Atypical Presentations of Leishmaniasis

Atypical manifestations of leishmaniasis are defined as cases from non-endemic areas or clinical presentations that are difficult to diagnose by clinicians and require the inclusivity of differential diagnosis. Unusual forms of leishmaniasis involve atypical causative species, rare morphological variants and unusual numbers and sites of lesions. Although the reasons for controlling the polymorphisms of leishmaniasis are not very clear, varied defence mechanisms of the host, the virulence of the parasite strain and immunosuppression are critical factors giving rise to the emergence of altered disease presentation in leishmaniasis. Other than HIV-infected situations, immunosuppressive drugs and immunomodulatory therapies are also risk factors for leishmaniasis severity [19]. Clinicians are well acquainted with the typical/usual clinical presentations of leishmaniasis. Unusual characteristics in morphology and localization go unrecognized. Compiled information about unusual presentations would be of great significance to medical knowledge for the clinical, diagnostic and disease therapeutics.

A comprehensive literature search was performed on the PubMed, Google and EuropePMC databases with the search terms “atypical leishmaniasis”, “unusual leishmaniasis”, “atypical visceral leishmaniasis”, “unusual visceral leishmaniasis”, “atypical post-kala azar dermal leishmaniasis”, “unusual post-kala azar dermal leishmaniasis”, “atypical cutaneous leishmaniasis”, “unusual cutaneous leishmaniasis”, “atypical mucocutaneous leishmaniasis” and “unusual mucocutaneous leishmaniasis”. We highlight the different atypical manifestations observed in patients of VL, PKDL, CL and MCL based on parameters such as unusual infective species, involved sites and characteristics based on morphology as well as present other associated factors, the role of the immune status of patients with such manifestations and the difficulty faced in the due course of treatment. Figure 1 depicts the geographical regions with incidences of atypical presentations observed globally.

### 2.1. Visceral Leishmaniasis

VL is characterized by “irregular bouts of fever, weight loss, enlargement of the spleen and liver, and anaemia” [5]. The amastigote forms spread to internal organs such as bone marrow, liver, spleen and lymph nodes through systemic circulation [20]. Untreated cases of VL lead to extensive wasting and bleeding due to thrombocytopenia. Furthermore, leukocytopenia leads to the suppression of the host’s immune system, leading to bacterial infections.

Atypical VL presentations in disease morphology and the localization of parasites among immunocompetent and immunocompromised individuals, particularly in the Indian sub-continent, are described herein. A VL suspect in a non-endemic region is infrequent and often misdiagnosed when the disease has an atypical presentation. However, a thorough investigation focusing on patient family records and travel history would enable VL diagnosis [21,22]. Travelling to endemic countries could cause the transmission of unusual pathogens, and VL should be considered in differential diagnosis when one a case presents with prolonged fever, hepatosplenomegaly and/or pancytopenia.

#### 2.1.1. Immunocompromised Cases with Unusual Presentation of VL

VL manifestations in immunocompromised individuals compared to immunocompetent patients are more atypical, often affecting gastrointestinal, pulmonary or laryngeal locations rather than bone marrow or spleen [23,24,25]. Compared to mono-infection, VL with co-infection often becomes challenging due to atypical presentation, poor prognosis, drug toxicity, resistance and early mortality [26]. VL and HIV mutually reinforce each other, as VL enhances the viral load and cytopenia and HIV infection leads to higher rates of treatment failure, relapse and death [26,27]. In suspected cases of glomerulonephritis, cryoglobulinemia was observed as a manifestation of VL [28]. Diffuse lymphadenopathy sustained by *L. infantum* is a complex clinical presentation classified as ‘‘paucisymptomatic leishmaniasis’’ with no visceral dissemination [29].

Cytomegalovirus (CMV) infection moderates the humoral and cell-mediated immune responses making the patient more susceptible to other infections. Many clinical features and hematologic parameters, including fever, leukopenia, and thrombocytopenia observed in CMV and leishmanial infection produce a diagnostic dilemma [27]. The negligible incidences of VL along with disseminated leishmaniasis similar to PKDL were also observed in Europe and South America [30]. An HIV-positive immunosuppressed regimen (Liposomal amphotericin B (LAmB), 5 mg/kg/day, 1 day per month for 6 months) was prescribed to all such cases.

#### 2.1.2. Unusual Presentation in Immuno-Competent Individuals

Some immunocompetent individuals present manifestations commonly seen in geriatric or immunocompromised individuals. The atypical presentation may involve the pulmonary or gastric system and occasionally the skin [31]. VL confined to the gastrointestinal tract in a young immunocompetent person had no other visceral affection, and diagnosis was based on a positive PCR and response to Liposomal amphotericin B (LAmB) therapy [32]. The duodenal invasion of the parasite causing the enlargement of the gastric walls prompted oesophagogastroscopy, which led to the identification of nonspecific duodenitis [33]. Morphological imaging commonly demonstrates hepatosplenomegaly in patients with VL; alternatively, the rarely reported fluorodeoxyglucose (FDG) avidity of the nodular splenic lesions may be helpful for the diagnosis of VL [34]. The patients were treated with LAmB.

#### 2.1.3. Leishman Donovan Bodies (LDBs) Localized in Unusual Body Parts

LDBs are localized in an infected individual’s spleen, liver and bone marrow. In rare cases, they may have unusual locations within myelocytes, plasma cells and megakaryocytes. In subclinical *L. infantum* infection, parasites were observed in the right adrenal gland where bone marrow aspirate diagnosis was a failure. The patient recovered well with the surgical removal of a mass with two internal cysts in the right adrenal region without antileishmanial treatment [35]. Acute acalculous cholecystitis (ACC), which causes the acute inflammation of the gallbladder despite the absence of stones, presented with splenomegaly and hepatomegaly; LDBs were cited in cystic lymph nodes on the gallbladder [36]. Oesophageal and laryngeal involvement in leishmaniasis has been reported in different regions of the world [37,38,39]. These cases were managed well with antimonials and antibiotics. Bilateral submandibular lymphadenopathy with hypopigmented lesions attributed to thrombocytopenia and platelet dysfunction is an uncommon VL manifestation [40].

#### 2.1.4. Miscellaneous Atypical Manifestations of VL

The human body becomes more susceptible to infectious diseases due to immunosenescence or immune ageing. VL is a rare and unusual diagnosis in nonagenarians. The case of febrile pancytopenia in a nonagenarian from the mediterranean coast of Spain diagnosed with VL was presented in [41]. VL is also an infrequent cause of the hemophagocytic syndrome (the uncontrolled amplification of hemophagocytic histiocytes [42]. Drug resistance or suboptimal treatment regimens may lead to the spontaneous reactivation of VL in immunocompetent individuals. Even after having anti-leishmanial treatment, disease relapse among children is generally associated with increased lethality rates, poor prognosis and disease severity, whilst multiple relapses enforce fatal co-infections. Misdiagnosis due to a low parasite count may often result in the inadequate administration of treatment dose, which leads to multiple episodes, as seen in a renal transplant patient [43]. However, in older age groups, age-induced immunosuppression underlines multiple relapses [44]. An HIV-positive immunosuppressed regimen was found to be more effective in controlling such relapse episodes [27].

### 2.2. Post-Kala-Azar Dermal Leishmaniasis

PKDL is a sequela of VL usually caused by *L. donovani*. The rash of PKDL usually appears as an asymptomatic combination of hypochromic macules, an erythematous papular rash or nodular or plaque-like lesions on the skin surface [45]. PKDL episodes are mainly reported in endemic areas of India, Bangladesh, Nepal and Sudan. After the apparent treatment of VL, the episodes of PKDL varies from 5% to 10%; occasionally, no history of past kala-azar is present [46,47]. In India, PKDL development after the successful treatment of VL could range from 6 months to 5 years or sometimes much longer. In contrast, PKDL is reported in approximately 56% of cured VL patients within weeks to a few months after treatment in Sudan [48]. Various factors, including cytokine levels in the host, drug dosage and irradiation under ultraviolet light may contribute to PKDL pathogenesis. VL is typified by the polyclonal B cell stimulation and Th2 immune response state with elevated levels of interleukin-10 (IL-10) and transforming growth factor (TGF)-β. After the treatment of VL, peripheral blood mononuclear cells (PBMCs) start producing IL-10, transforming from a T-helper type 2 (Th2) to T-helper type 1 (Th1) or a mixed Th2/Th1 immune response resulting in PKDL. Increased levels of CD8 T cells before treatment were implicated to cause skin ulceration [49].

#### Atypical Presentations of PKDL

The polymorphic manifestation of PKDL may uncommonly be limited to a predominantly monomorphic presentation such as macular, papular or papulonodular forms. Rare morphological forms of PKDL, either localized or disseminated including mucosal, xanthomatous, verrucous, papillomatous, hypertrophic, fibroid, atrophic and extensive tumorous or hypopigmented monomorphic forms, have been documented in regions endemic to PKDL [50,51]. Differentiating even the common presentation of PKDL from leprosy is a recurring problem in co-endemic areas [52]. Though very uncommon in India, lymph node and nerve involvement without impaired sensation in PKDL has been reported in Sudan. Leprosy shows loss of sensation, motor weakness, nerve enlargement, and acid-fast bacilli in slit-skin smears. However, tuberculoid leprosy can be differentiated on histological grounds where nerve fibres are completely absent or only present within granulomas and not in between them [53]. An unusual combination of healed leprosy sequelae and active PKDL lesions was reported in Bihar, India [54].

In endemic areas of PKDL, the nodules involving mucosa may appear in the corners of the mouth, dorsum of the tongue, buccal mucosa or soft palate, sometimes involving the upper respiratory tract leading to ulcerations [45,55]. Hypopigmented macules, coalescing to form patches on the trunk, caused the misled diagnosis of pityriasis versicolor. Mucosal PKDL with polymorphic skin lesions and gradually developing hypopigmented patches were reported in a patient who developed erythematous papular and nodular lesions over the face, neck and trunk, along with the concomitant involvement of perioral mucosa and tongue (Figure 2a) [56].

Polymorphic PKDL involving an oral cavity exhibiting areas of melanin pigmentation interspersed with fine, white radiating striae on the right buccal vestibule and the buccal mucosa has been reported (Figure 2b) [57]. PKDL may also manifest atypically as the recurrent swelling of the muzzle area (Figure 2c) of the face (area pertaining to the primate muzzle area of the face), namely the lips and perioral area extending onto the cheeks on both sides [58]. The disseminated annular lesions of PKDL, which were skin-coloured or mildly erythematous and that had indurated annular plaques with central clearing, irregular in shape, and had soft and non-tender papules on the face, hands, back and thighs with a histology closely mimicking that of granuloma annulare were reported (Figure 2d) [59].

Nodular lesions on the mucosa of the glans penis, anus and oral cavity up to the vocal cord have been documented in PKDL patients [63]. In Indian PKDL, ulcerated lesions are rare, and even when present they can be attributed to trauma as seen in the tumorous, eroded and non-tender plaque on the forehead of an adult male who, when praying, used to strike the ground with his forehead (Figure 2e), or in the person with an ulcerated nodule on the dorsum of the foot most likely caused by repeated trivial trauma [64]. Even in such cases of PKDL, multiple asymptomatic, hypopigmented patches over the face and trunk that diffusely infiltrate have been described (Figure 2f) [60,61,65].

Disease over a long duration has a devastating impact in terms of economic activity, social stigmatization and isolation. Prolonged disease may result in organ disfigurement (Figure 2g), visual impairment and mental distress [62]. Another case of delayed treatment but having experienced significant healing with miltefosine treatment was reported in ulcerated PKDL (Figure 2h) [49]. Ocular leishmaniasis caused by *L. donovani* [66,67] or the dermotropic *Leishmania* spp. is occasionally reported [68,69]. It has diagnostic limitations as leishmanial infection is hardly suspected of causing eye lesions and parasite demonstration is very difficult [70]. Ocular leishmaniasis requires early diagnosis and treatment to prevent permanent damage to the eyes. It could develop as post-kala-azar leishmanial conjunctivitis and blepharitis or post-kala-azar anterior uveitis with red conjunctivae, marked ciliary injection, oedematous corneal epithelia and pigmented precipitates in the corneal endothelia [66]. In OW regions *L. major* infection causes cutaneous disease, while conjunctivitis and chalazion-like lesions are rare [69]. A change in immune response from a Th2 to a combined Th1/Th2 pattern underlines ocular leishmaniasis after VL treatment resulting in blepharo-conjunctivitis or pan-uveitis [71]. In HIV–VL co-infected patients, ocular leishmaniasis is postulated to be a part of highly active antiretroviral therapy (HAART)-induced immune reconstitution syndrome [72]. Reports of ocular disorders including rare cases of unilateral and bilateral blindness, permanent in some cases, ulcerative keratitis, leukocoria, blurred vision, ocular hyperaemia, photophobia and eye pain, in patients treated with miltefosine for PKDL have mostly originated from India [73]. In several cases, as reported, the symptoms resolved following the discontinuation of the treatment, indicating it could possibly be due to miltefosine.

‘Para-kala-azar dermal leishmaniasis’ is the term used when a patient develops PKDL during treatment for VL. The co-occurrence of VL and PKDL has been commonly reported in East Africa, however, it is rare in the Indian subcontinent where only isolated cases are reported, and such presentations due to ineffective immune response are common in HIV–VL co-infected patients. Immuno-compromised individuals are prone to frequent relapses and can manifest a variety of co-infections [45]. An HIV–VL co-infected patient developed PKDL during the LAmB treatment of VL by *L. infantum* [74]. PKDL with *L. infantum* can develop mucocutaneous lesions highly suggestive of Kaposi’s sarcoma (KS) or clinically manifest as immune reconstitution inflammatory syndrome (IRIS). In an asymptomatic papular PKDL case, a rash over the torso, arms, thighs and face in a patient with HIV and cerebral toxoplasmosis co infection was described [75].

### 2.3. Cutaneous Leishmaniasis

Localized CL (LCL) is regarded as the most common type of leishmaniasis. CL lesions can persist for several months, and in a few cases, even years. CL lesions typically “evolve from papules to nodular plaques to ulcerative lesions, with a raised border and central depression, which can be covered by scab or crust; some lesions persist as nodule.” [9]. Generally, these lesions are painless, however, they may be painful, which sometimes pertains to their presence near joints or due to bacterial infection. The size and appearance of CL lesions may change over time.

#### 2.3.1. Unusual Presentations Caused by Atypical Species

Several reports present atypical cutaneous leishmaniasis in terms of infecting species, e.g., the involvement of VL, which causes *Leishmania* species in causing cutaneous manifestations and vice versa [76]. Species determination plays a critical role in understanding the presentation of the disease and deciding the treatment that the patient should undergo during the initial phases of diagnosis. Furthermore, if *L. donovani* or *L. infantum* are identified as the causative species for CL, it is necessary to rule out the involvement of visceral organs.

*L. donovani* causing CL has been reported in imported cases in the United Kingdom, regions of the Mediterranean basin, Brazil, the Western Ghats of India and Sri Lanka [77,78,79,80]. DNA sequencing and microsatellite analysis has shown that *L. donovani* is a causative species for CL in Sri Lanka and belongs to zymodeme MON-37, which closely resembles the spp. from the ISC MON-2 zymodeme which has been reported in various regions of the world such as Ethiopia, India, Israel and Turkey, which is responsible for causing VL [78,81,82]. *L. donovani* strains causing VL and CL in India exhibited a difference in GPI and gp63 sequences [83]. Researchers have emphasized the differences in MON-37 isolated from Cyprus and India, Sri Lanka, Israel, Kenya and Turkey [84,85,86]. Genetic variations, alterations in immune and treatment responses might lead to atypical CL. *L. donovani* strains causing CL in Sri Lanka were found to contain a gene polymorphism homologous to *L. major* and *L. tropica* genomes and were quite distinct from *L. donovani*, causing VL in India. Furthermore, intraspecies hybridization in *L. donovani* has been demonstrated to cause a phenotypic reposition from visceral to cutaneous form, causing an atypical CL phenotype. The recombinant derived from a CL focus in Himachal Pradesh resulted from genomic hybridization between the two parental types of *L. donovani*, which belong to the Yeti ISC1 variant of the Nepalese highlands [87]. In atypical CL cases due to *L. donovani* in Sri Lanka, sodium stibogluconate is the opted treatment.

CL due to *L. infantum* was reported in the Mediterranean region [88,89], Morocco [90,91], different countries of North Africa [92], Tunisia [93,94,95], France [96], New York City (United States) [97], Turkey [98,99], Iran [100], Spain [101,102,103], and Brazil [104,105]. *L. infantum* and *L. donovani* were reported as infective species for CL in northwestern Yemen, Turkey, Syria, Iran, Lebanon and Southern Israel, where *L. tropica* and *L. major* are predominant CL-causing species [100,106,107,108,109]. CL lesions due to *L. donovani* are more severe than those due to *L. infantum* and the response to antimonials is often different. Cases of CL with a single crusted lesion due to *L. (Mundinia) orientalis* were described in patients from Thailand [110,111]. LAmB has been opted as a drug of choice in CL cases due to *L. donovani* as well as *L. (Mundinia) orientalis* [110,112].

#### 2.3.2. Unusual Sites and Number of Lesions

The lesion in the submandibular region is a rare clinical variant of CL and might mimic a parotid neoplasm [113] and may be devoid of ulcer or crust [114].

Lip leishmaniasis, a relatively new variant of CL, is occasionally reported from India, Saudi Arabia and Turkey [115,116,117]. It proceeds with the ulceration of both upper and lower lips, with macrocheilitis as the ultimate clinical presentation [118]. The infective species were identified as *L. tropica* (which rarely involves the lips) and *L. major* through PCR in lip leishmaniasis lesions (Figure 3a) [119,120]. Failure in skin smear and biopsy examination may lead to difficulty in the diagnosis [121]. Biopsy examination revealing amastigotes proves to be helpful when lesions indicative of CL are absent [118]. Primary lip leishmaniasis from a non-endemic area of the Kashmir valley, India, has been described.

All the patients reported an indurated crusted nodule on their lip as the primary lesion. Serous discharge was observed in two cases, while bleeding occurred in the other two [124]. Although leishmaniasis recidiva cutis (LRC) has been reported to some extent, the LRC of lips is a very rare presentation of CL in terms of the site of the lesion. LRC of lips, resembling granulomatous cheilitis, is an unusual form of CL occurring in NW regions [125]. The differential diagnosis for lip leishmaniasis should include diseases such as herpes labialis, orofacial granulomatosis, oral Crohn’s disease, syphilitic chancre, cutaneous tuberculosis, lymphoma and carcinoma [120].

Auricular leishmaniasis is a rare variant of CL since the auricle of the ear is a rare site of infection in OW leishmaniasis. In NW regions, it is commonly found as a chiclero’s ulcer affecting the ear’s pinna. It develops upon infection with *L. mexicana* in forest workers harvesting chicle gum from plants [126]. Occasional cases involving different parts of the ear have been described where the lesion mimicked neoplastic disease progression [127]. The lesions of auricular leishmaniasis mimicking squamous cell carcinoma were described [128,129,130]. *L. tropica* is an atypical species causing chiclero’s ulcer (Figure 3b) [122,131]. The differential diagnosis for auricular leishmaniasis includes the symptoms of infections such as lupus vulgaris, mycobacterial disease, fungal infections and syphilis with the involvement of the ear as a site, and which may lead to the misdiagnosis of CL [132].

Ocular leishmaniasis, an uncommon variant of CL, may lead to irreparable damage caused by ophthalmologic complications, sometimes resulting in blindness [133]. The ulceration of the eyelids causes keratopathy. Granulomatous uveitis, interstitial keratitis and phlyctenulosis may also develop as complications of eyelid ulceration [134]. *L. infantum*, *L. major* and *L. tropica* were the infective species in various cases of ocular leishmaniasis (Figure 3c) [123]. Incidences of ocular leishmaniasis in South America due to *L. braziliensis* were reported [135]. Lesions could develop on the upper or lower eyelid or on eyelid margin as an erythematous nodule with indurated edges [136,137,138,139,140,141,142,143]. The involvement of the conjunctiva can be a severe problem and may lead to blepharoconjunctivitis [144,145]. Chronic and erosive ulcerations are suspected to be capable of leading to the development of diseases including chalazion, tuberculosis, syphilis, sarcoidosis and basal cell carcinoma [146]. The differential diagnosis for ocular leishmaniasis includes pterygium, chronic blepharitis, cell carcinoma, impetigo and histoplasmosis [147]. The haired and bald scalp and palm are other unusual sites that can be affected by CL [148,149]. Differential diagnosis of scalp CL includes kerion celsi, dermatitis and psoriasis.

The involvement of genitals in CL is uncommon. Leishmaniasis of the penis is represented by nodules, papules, erythematous ulcerations and crust. CL lesions on the glans penis and scrotum are generally painless and could be ulcerative [150,151,152,153]. The diagnosis of leishmaniasis was made by histopathological examination and/or PCR and the preferred treatment was cryotherapy [154,155]. Differential diagnosis includes squamous cell carcinoma, basal cell carcinoma, adenocarcinoma, lymphoma and viral infections such as human papillomavirus (HPV), human immunodeficiency virus (HIV), syphilis, fungal infections, bacterial infections and psoriasis [154,155].

CL lesions are commonly 1–5 in number and present on exposed body parts. A large number of lesions is an unusual phenomenon [123,154,156,157].

#### 2.3.3. Unusual/Atypical Morphology/Characteristics

Different types of morphological variants of CL lesions are presented in Figure 4a–f and have recently been reviewed in detail [158,159]. The erysipeloid form, a rare variant often leading to late diagnosis, could be present in the form of an erythematous indurated plaque [160], a small ulcerative and cribriform lesion or a papule with redness and induration [161,162]. The erysipeloid form, in which ulcerative erythematous lesions develop, has been previously reported in Pakistan, Nepal, Iran and Turkey [157,162,163,164,165,166,167,168,169]. In general, the lesion responds well to antimony treatment [168,170]. A large erythematous plaque studded with papules and pseudo-vesicles developed in zosteriform such as CL resembles the lesions caused due to herpes zoster infection [171]. Chronic zosteriform CL in covered body parts is a rare clinical presentation [172].

Sporotrichoid leishmaniasis (SL) is an unusual form described in combination with the inflammation of the lymphatic system. The symptoms of sporotrichosis may complicate the differentiation between SL and sporotrichosis [173,174,175]. SL is reported to be prevalent in females compared to males, with SL lesions predominantly in upper limbs [173]. CL with mucosal involvement presenting with multiple papules and nodules in line with the sporotrichoid form of leishmaniasis as well as hepatosplenomegaly along with HIV co-infection due to *L. (Mundinia) martiniquensis* was observed in Thailand [15].

Lupoid-form CL or cutaneous lupoid leishmaniasis (LL) develops with a distinctive expansion of the primary lesion into erythematous granulomatous infiltrative plaques with superficial desquamation in a butterfly-like pattern. The lesions may sometimes combine, and the resultant plaque closely resembles lupus vulgaris.

Although not as destructive as lupus vulgaris, LL may persist and spread for many years [176]. In contrast to leishmaniasis recidivans (LR), which is a recurrently appearing lesion, LL is a rare clinical variant of CL with a peripheral lupoid type spread. Granulomatous papules developed in LL might be associated with ulceration and crusting [177]. LL is most prevalent in the region of the Middle East and approximately 4% of CL cases presented lupoid leishmaniasis [117,178]. Differential diagnosis includes lupus vulgaris, discoid lupus erythematous, lupus pernio and erysipelas. The face is reported as the most common site of LL lesions [179,180,181]. A few cases of CL with a clinical spectrum suggestive of LL have been reported in the non-endemic areas of Rajasthan, India, without any prior travel history to endemic regions of CL [182].

Leishmaniasis recidivans (LR) in OW or leishmaniasis recidiva cutis (LRC) in NW regions is an uncommon variant of CL [157,183], with the development of red-brown papules developed on the exposed body parts. Generally recurring at the site of the primary ulcer, it often does not respond to treatment [184]. It is caused by *L. tropica* in OW regions, whereas in new world species, *L. amazonensis*, *L. guyanensis* and *L. panamensis* are responsible [185]. LR/LRC is regarded as the reactivation of dormant parasites after a considerable period leading to the recurrence of lesions at or around the site of a previous acute lesion [183]. The period for the recurrence of LR could be as long as 43 years [186].

Disseminated cutaneous leishmaniasis (DCL) is an uncommon pattern of CL with the development of several lesions, variable in shape and size on nonadjacent body parts. Disseminated maculopapular rashes may develop following VL relapse in an immunocompromised patient [187]. Such clinical presentation poses difficulty in the disease diagnosis. DCL is marked by the unrestricted development of infection, unresponsive cell-mediated immunity and a lack of response to treatment. The autochthonous infection of DCL caused by *L. amazonensis* was reported from Brazil [188]. Disseminated leishmaniasis due to *L. (Mundinia) martiniquensis* with chronic fibrotic lesions covering most parts of the body along with VL and HIV co-infection from northern Thailand were reported [189]. A case of autochthonous disseminated dermal leishmaniasis due to *L. (Mundinia) martiniquensis* from Myanmar was described wherein the patient developed numerous erythematous nodules on face, trunk and extremities [190]. The patients with *L. (Mundinia) martiniquensis* were treated with intravenous amphotericin B [189,190].

### 2.4. Mucocutaneous Leishmaniasis (MCL)

MCL, also known as ‘espundia’, is “metastatic sequela of New World cutaneous infection, which results from dissemination of parasites from the skin to the naso-oropharyngeal mucosa”. MCL progresses in cases where the treatment of CL lesions was undermined or was not treated optimally. Generally initiated with prolonged nasal issues such as bleeding or stuffiness, it progresses into the degeneration of nasopharyngeal mucosa, ultimately leading to the destruction of the nasal septum in patients [9]. The degeneration of the nasal septum and nasopharyngeal mucosa impairs the function of the nose.

#### 2.4.1. Unusual/Atypical Variants of Causative Species

MCL is sporadically reported in OW regions [191]. Many previous reports of mucosal presentations due to *L. donovani* and *L. infantum* in OW regions have been collated (reviewed in [192]). Sudanese MCL and classical VL, caused by *L. donovani*, can be differentiated with the former possessing characteristics resembling those of *L. major* [193,194]. Lesions on the perioral mucosa sublingual space, gingiva and palate have been attributed to MCL due to *L. donovani* in patients from India, Sri Lanka and Malta [195,196,197,198]. Other complications are recurrent epistaxis, nasal obstruction and granular lesions on the uvula, soft palate and tonsils [199].

MCL due to *L. infantum* is reported from Spain [192], Tunisia [200], France [201] and Italy [202]. A rare presentation of local MCL in the laryngeal region in an immunocompetent patient where white lesions were present on the epiglottis was reported from Italy [203]. *L. infantum*, *L. major* and *L. tropica* are the major causative species of oro-mucosal leishmaniasis in Iran [204]. The co-infection of *L. major* and *L. tropica* can cause nasal and mucosal lesions in a patient [205,206]. Isolated lingual leishmaniasis due to either *L. major* or *L. tropica* was also reported [207]. A correlation between the species and type of lesion observed in patients was outlined where *L. major* was associated with lesions on the palate (hard and soft), gingival and nasal tissues, and *L. tropica* with lesions on the gingiva and lower lip, *L. infantum* with the epiglottis and laryngeal mucosal part [204].

#### 2.4.2. Unusual Sites/Morphology/Characteristics

Rare presentations such as ocular scleromalacia may develop as a complication of MCL [208]. A disseminated form of MCL is also rarely observed [209]. Other reported unusual observations are primary endonasal leishmaniasis [210] and focal hard whitish lesions on true vocal cords (Figure 5a) [202]. The exclusive involvement of oral mucosa is rare [211].

The spontaneous healing of MCL in HIV-infected patients is rarely observed; however, patients may respond to the treatment, possibly due to an enhanced immune status pertaining to HAART [212]. The localized MCL of the oral mucosa (Figure 5b) is rare [213]. Oral leishmaniasis with the primary lesion (erythematous and oedematous) and without the involvement of cutaneous tissue [214] or an ulcer with a punched-out appearance extending to the lower lip and associated with the oedema of oral mucosa resembling neoplasm is unusual [215].

The involvement of oro-facial mucosa [216], nasal mucosa, uvula and pharyngeal mucosa and cartilage bone septum [217] are also rare presentations. A relapse of MCL may progress into DCL, with no response to treatment for both SSG mono and SSG and paromomycin combined therapy [218]. MCL and HIV co-infection with desquamative rashes and erythematous and nonpruritic lesions are less frequently observed [219]. The smooth and erythematous swelling of the lower lip (chelitis) and granulomatous disease of the endolarynx [220] as well as lesions on the conjunctiva of the upper and lower eyelid [221] (Figure 5c) are some other unusual morphological variants of MCL. Oropharyngeal mucosal leishmaniasis (Figure 5d) is also a rare presentation [222]. Sublingual leishmaniasis with atypical pseudotumoral morphology [223] and lingual leishmaniasis (Figure 5e) with lymphoid-like tissue swelling on the dorsal part of the tongue [224] are some of the rare observations in MCL. The differential diagnosis of the mucosal form includes neoplasm, tuberculosis and squamous cell carcinoma. Antimonials and LAmB are the drugs of choice to which patients have responded well in atypical cases [112].

Photodynamic therapy has been opted as a treatment option [122]. Cryotherapy [225] and excisional biopsy are other methods adopted for the treatment of localized lesions [166].

**Figure 5 pathogens-12-00297-f005:**
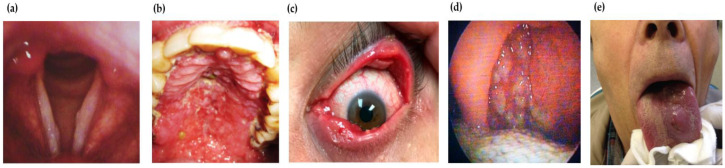
Atypical sites/locations of MCL lesions: (**a**) focal hard whitish lesions on the true vocal cords of a patient with MCL [202]; (**b**) lesions on hard and soft palate and cobblestone appearance of lesions on hard palate of MCL patient [213]; (**c**) lesions on conjunctiva of the upper and lower inner eyelids [221]; (**d**) multiple granulomatous MCL lesions on the posterior wall of the oropharynx [222]; and (**e**) lingual leishmaniasis showing swelling on the dorsal part of lingual tissue [224].

## 3. Coinfection and Unusual Presentation

The presence of a co-infection/morbidity and leishmaniasis affects the host’s immune system to a great extent. The high prevalence of malaria in VL infected individuals is due to the overlap of spatial niches of *Leishmania* and *Plasmodium*, and *Leishmania* impairing the host’s immune system [226]. Co-infection of leishmaniasis and schistosomiasis occurs due to their wide distribution in tropical regions [227]. *Leptomonas seymouri* is another opportunistic trypanosomatid infecting patients with VL, PKDL and CL patients [228,229]. Differentiation between sporotrichosis and sporotrichoid CL is difficult as the latter develops a subcutaneous nodule similar to sporotrichosis. The clinical and epidemiological characteristics of both are identical, raising concerns during diagnosis [158]. Co-infection with sandfly fever Sicilian virus exacerbates CL lesions [230]. Leishmaniasis acts as an opportunistic disease in human immunodeficiency virus (HIV)-infected patients and leads to the progression of HIV-specific clinical features affecting the life expectancy of individuals. *Leishmania* takes advantage of the suppressed immune system, thus comprehensively causing exaggerated clinical symptoms [231,232].

## 4. Conclusions

Distributed in the tropics and subtropics, leishmaniasis globally affects millions of people. Typical presentations of the different manifestations of leishmaniasis have been well delineated by the WHO and facilitate correct clinical diagnosis. However, incidences of atypical forms of the disease in all four types pose a considerable challenge for clinicians during the initial screening of leishmaniasis. We included as many and as diverse reports of atypical/unusual findings in leishmaniasis in this review as possible; however, we may have missed a few. Unusual manifestations of leishmaniasis could be due to co-morbidities/co-infections, unexpected transition during treatment, nourishment status and immunosuppression in individuals (Table 2). Clinical presentations recorded in immunocompromised patients are more atypical compared to immunocompetent patients and are challenging in terms of diagnosis and treatment. Case management and control require a clear understanding of the characteristics noted during disease presentation and any sequela that might develop post-treatment and the aetiology of clinical variants. Pentavalent antimonials and liposomal amphotericin B were the most preferred drugs to control cases complicated with atypical characteristics. The determination of infective species and inclusion of rare variants in differential diagnosis will aid in the timely diagnosis and treatment of leishmaniasis.

## Figures and Tables

**Figure 1 pathogens-12-00297-f001:**
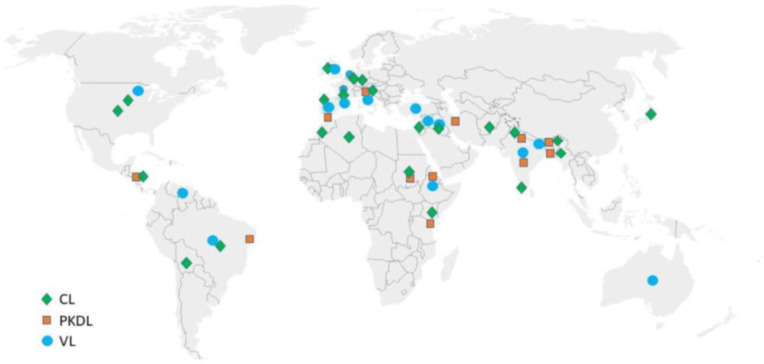
Geographical localization of atypical presentations observed in Leishmaniasis. CL: Cutaneous leishmaniasis, PKDL: Post kala-azar dermal leishmaniasis, VL: Visceral leishmaniasis.

**Figure 2 pathogens-12-00297-f002:**
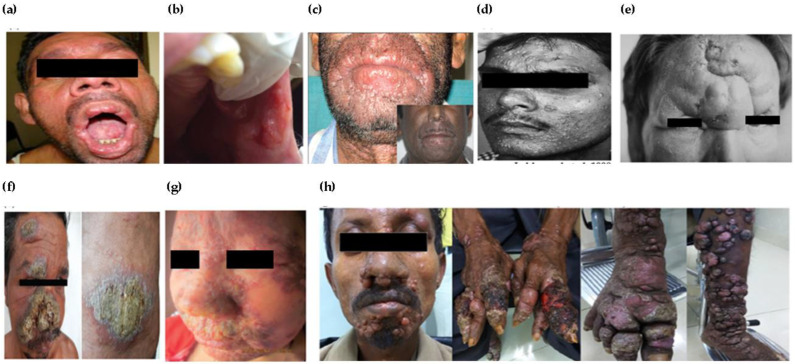
Atypical sites/locations of PKDL lesions: (**a**) unusual involvement of perioral and lingual mucosa [56]; (**b**) erythematous mucosa with ulcer on retro-commissural area [57]; (**c**) swelling of lips and muzzle area in patient, (inset: post treatment) [58]; (**d**) annular lesions on the face of the patient [59]; (**e**) unusual swelling on forehead in PKDL patient [60]; (**f**) crusted ulcerative plaques on face and abdomen [61]; (**g**) crusted erythematous plaque along with loss of vision [62]; (**h**) severely ulcerated papulo-nodular lesions on face and limbs of patient [49].

**Figure 3 pathogens-12-00297-f003:**
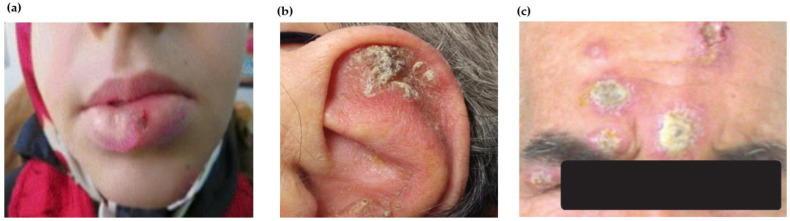
Atypical site/location of CL lesions: (**a**) lip leishmaniasis in patient with red ulcer and crust covering acute swelling on the lower lip [120]; (**b**) auricular leishmaniasis in patient with indistinct plaque with hyperkeratosis on the antihelix of the ear [122]; (**c**) ocular leishmaniasis in a patient with several lesions on the face and on right upper eyelid [123].

**Figure 4 pathogens-12-00297-f004:**
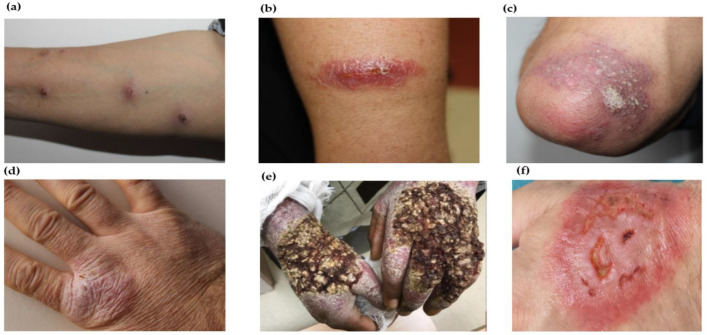
Atypical morphology of CL lesions: (**a**) sporotrichoid-type lesions on the upper extremity of the CL patient; (**b**) lupoid form of lesions observed in arm of the CL patient; (**c**) psorasiform type of CL lesion on the elbow of the patient’s right arm; (**d**) eczematoid-type CL lesion on back of the hand; (**e**) verrucous-type CL lesion on back of the hand; 4 (**a**–**e**). [158]; and (**f**) multiple erythematous lesions with indurated margin and necrotic-purulent base mimicking pyoderma gangrenosum [159].

**Table 1 pathogens-12-00297-t001:** Leishmaniasis manifestation and infective spp.

Manifestation	Presentation	Species
Visceral leishmaniasis	Typical	*L. donovani*
		*L. infantum*
		*L. chagasi* *L. (Mundinia) martiniquensis*
	Atypical	*L. tropica*
Cutaneous leishmaniasis	Typical	*L. major*
		*L. tropica*
		*L. mexicana*
		*L. amazonensis*
		*L. braziliensis* *L. (Mundinia) orientalis*
	Atypical	*L. donovani*
		*L. infantum*
Mucocutaneous leishmaniasis	Typical	*L. amazonensis*
		*L. braziliensis*
		*L. panamensis*
		*L. guyanensis*
	Atypical	*L. donovani*
		*L. infantum*
		*L. aethiopica*
		*L. major*
		*L. tropica*

**Table 2 pathogens-12-00297-t002:** Unusual observations in various types of leishmaniasis and possible causes.

Disease Manifestations	Typical Manifestation	Unusual Observations	Plausible Mechanisms
Visceral leishmaniasis (VL)	Irregular bouts of fever, weight loss, anaemia, enlargement of spleen and liver LDB (amastigotes) spread to internal organs such as bone marrow, liver, spleen and lymph nodes through systemic circulation	Gastrointestinal tract, pulmonary system, larynx and skin are involved in addition to liver, spleen and bone marrow in in both immunocompromised and immune-competent cases LDB (amastigotes) seen in myelocytes, plasma cells and adrenal gland	Unusual presentation of an outcome of compromised immune response due to intrinsic poor immune system, co-infection with HIV or other pathogens Immune senescence leads to VL in geriatric population resulting in multiple relapses
Post-kala-azar dermal leishmaniasis (PKDL)	Dermal sequela of VL usually caused by *L. donovani* as a polymorphic presentation of macular, papular or nodular rash on face, upper arms, trunks and other parts of the body	Monomorphic presentation including macular, papular or papulonodular forms Localized or disseminated including mucosal, xanthomatous, verrucous, papillomatous, hypertrophic, fibroid, atrophic and extensive tumorous forms Lymph node and nerve involvement without impaired sensation Mucosa of genitalia, anus, lingual, perioral and oral cavity involved Indurated annular plaques with central clearing, irregular in shape, soft and non-tender juicy-looking papules, ulcerated lesions as seen in the tumorous, eroded and non-tender plaque Ocular leishmaniasis caused by *L. donovani* or dermotropic spp. causing permanent damage to the eyes, developed as post-kala-azar leishmanial conjunctivitis and blepharitis or post-kala-azar anterior uveitis	Ulcerations possibly due to repeated trauma Change in immune response from Th2 to a combined Th1/Th2 pattern underlines ocular leishmaniasis Antiretroviral therapy induced immune reconstitution syndrome among HIV–VL co-infected patients
Cutaneous leishmaniasis (CL)	Localized lesions at site of bite with changing appearance and size with course of time Mostly painless, however, may be painful pertaining to their presence near joints or due to bacterial infection	Atypical in terms of infecting species, e.g., VL-causing *leishmania* species causing cutaneous manifestations and vice versa CL at unusual sites including lesions in submandibular region mimic parotid neoplasm, auricle of ear, eyelids, haired and bald scalp, palm or lips, genitals (glans penis, scrotum) Morphological variants of CL lesions including erysipeloid form; chronic zosteriform CL in covered body parts; sporotrichoid form predominantly in upper limbs; lupoid form; leishmaniasis recidivans in the Old World regions or leishmaniasis recidiva cutis in New World regions Disseminated maculopapular rashes post relapse of VL	Genetic variations including gene polymorphisms or intra-species hybridization Alterations in immune response Treatment responses
Mucocutaneous leishmaniasis (MCL)	Metastatic sequela of New World cutaneous infection Dissemination of parasites from the skin to the naso- oropharyngeal mucosa causing degeneration and ultimately leading to destruction of the nasal septum	Oral leishmaniasis with the primary lesion (erythematous and oedematous) without involvement of cutaneous tissue Lesions on perioral mucosa, oro-facial mucosa, nasal mucosa, pharyngeal mucosa and cartilage bone septum, uvula, gingiva, soft palate, tonsils and epiglottis Recurrent epistaxis or nasal obstruction Primary endonasal leishmaniasis, focal hard whitish lesions on true vocal cords Ulcer with punched-out appearance extending to lip, smooth and erythematous swelling of lips (chelitis), granulomatous disease of endolarynx and oedema of oral mucosa resembling neoplasm Sublingual leishmaniasis with pseudotumoral morphology, lingual leishmaniasis with lymphoid-like tissue swelling Ocular scleromalacia, lesions on the conjunctiva of upper and lower eyelid, Disseminated MCL	Inadequate treatment of CL lesions Co-infection with interspecies strains

## Data Availability

Not applicable.

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
