# Peer review of "Unusual Observations in Leishmaniasis—An Overview"

_pathogens, 2023, doi:10.3390/pathogens12020297_

Round 1

Reviewer 1 Report (Previous Reviewer 2)

Thank you for your response. 

I have read your manuscript carefully and I appreciate the vast work you have done. With the changes introduced, the classification is much clearer. However, in my opinion there is too much information that makes it hard to read and to extract concrete messages. I would suggest summarizing cases in a table and give brief explanations to the underlying mechanisms in such form (visceral, cutaneous, etc.). I would also insist in my recommendation of shortening the introduction. In my opinion, as it is currently written, makes it unsuitable for publication in this journal.

Author Response

Reviewer 1 I have read your manuscript carefully and I appreciate the vast work you have done. With the changes introduced, the classification is much clearer. However, in my opinion there is too much information that makes it hard to read and to extract concrete messages. I would suggest summarizing cases in a table and give brief explanations to the underlying mechanisms in such form (visceral, cutaneous, etc.). I would also insist in my recommendation of shortening the introduction. In my opinion, as it is currently written, makes it unsuitable for publication in this journal.

Response: As recommended, the introduction is shortened now by omitting the history and erstwhile existence. However, as another reviewer recommended, information on Leishmania's taxonomic position is added in the introduction. A brief table, as suggested, has now been added.

Reviewer 2 Report (Previous Reviewer 1)

Dear authors,

I have enjoyed reading the submitted manuscript and consider it a significant scientific contribution. Here are some observations and minor changes that should be considered.

1. Add % information for MCL and the reference. The same information is needed for PKDL (Lanes 39-40).

2. Table 1. Do not use capital letters for the diseases (i.e., replace: VISCERAL LEISHMANIASIS for Visceral leishmaniasis).

3. Lane 144, Reference for Figure 1, the figure was adapted from a previous reference (if so, the authors request authorization to use the image?), is a new figure based on the previous study? Was it generated with which information? Is the information presented supported by the NIH, and CDC databases? Finally, which program was used to create the figure, and how accurate is it?

4. Figure 1. It needs to be improved, referenced, and clarified if it is only related to “atypical leishmaniasis.”

5. Add references on:

Lanes 171-173 (ref).

Lanes 202-204 (ref).

Lanes 204-205 (ref).

6. Figures 2-5. The authors need consent to use the images from the original authors (which is not described or mentioned in the manuscript, please request permission).

The copyright owner controls the right to publish a copyrighted image, so each copyrighted image that you use must have permission or fail within an exception to the general copyright statute, such as public domain, fair use, or open access.

7. Remove the gap on lanes 338-339.

8. Update and homogenize references. Sometimes the authors used a gap between the text and the reference number (in all the manuscripts).

9. Add reference 127 on Lane 379.

Author Response

I have enjoyed reading the submitted manuscript and consider it a significant scientific contribution. Here are some observations and minor changes that should be considered.

  1. Add % information for MCL and the reference. The same information is needed for PKDL (Lanes 39-40).

Response: The information is added at the designated place.

  1. Table 1. Do not use capital letters for the diseases (i.e., replace: VISCERAL LEISHMANIASIS for Visceral leishmaniasis). 

Response: Corrected.

  1. Lane 144, Reference for Figure 1, the figure was adapted from a previous reference (if so, the authors request authorization to use the image?), is a new figure based on the previous study? Was it generated with which information? Is the information presented supported by the NIH, and CDC databases? Finally, which program was used to create the figure, and how accurate is it?

Response: The figure generated showing regions for atypical presentations of VL, PKDL, CL have been marked on the map based on the studies covered in this review. The figure was generated manually (Microsoft ppt) by marking the location of cases in the world map.

  1. Figure 1. It needs to be improved, referenced, and clarified if it is only related to “atypical leishmaniasis.” 

Response: The figure generated showing regions for atypical presentations of VL, PKDL, CL have been marked on the map based on the studies covered in this review. Marking references of all the cases reported will make it crowded.

  1. Add references on:

Lanes 171-173 (ref)-  Ref added.

Lanes 202-204 (ref). and Lanes 204-205 (ref).

Response: Same reference for both lanes was added.

  1. Figures 2-5. The authors need consent to use the images from the original authors (which is not described or mentioned in the manuscript, please request permission).  The copyright owner controls the right to publish a copyrighted image, so each copyrighted image that you use must have permission or fail within an exception to the general copyright statute, such as public domain, fair use, or open access.

Response: The images are mostly from open-access articles. A file indicating the permission is attached separately.

  1. Remove the gap on lanes 338-339.

Response: Complied.

  1. Update and homogenize references. Sometimes the authors used a gap between the text and the reference number (in all the manuscripts).

Response: Corrected.

  1. Add reference 127 on Lane 379.

Response: Added.

Reviewer 3 Report (New Reviewer)

To authors,

The manuscript has been clearly described regarding unusual presentations of leishmaniasis previously reported from different geographical regions. As mentioned in the review, the authors stated that Leishmania were classified in two subgenera, namely Leishmania and Viannia, and it appears that the authors would focus unusual manifestation resulting from members of such two subgenera, especially in ISC countries. However, the update classification of Leishamania parasite reveals a total of four subgenera, including Leishmania, Viannia, Sauroleishmania (reptile-infecting), and recently emerging Mundinia. The subgenus Mundinia is previously recognized as 'L. enriettii' complex and consists of five members, namely L. martiniquensis, L. orientalis, L. sp. Ghana, L. macropodum, and L. enriettii. To make your paper perfect, I would suggest that the authors should update the taxonomic information and the map of geographical distribution and include more relevant references especially of VL-causing Mundinia species, L. martiniquensis (formerly known 'L. siamensis').

Of interest, the Mundinia members have been continuously reported with increasing frequency, in Southeast Asian countries, especially Thailand and Myanmar. Of interest, a variety of unusual manifestations due to L. martiniquensis in immunosuppressive hosts (HIV and systemic steroid therapy), have been formerly published. I would suggest you include more interesting clinical data as revealed in the following references:

1. Disseminated erythematous, shiny infiltrative plaques and nodule on face, trunk, and extremities:

Noppakun et al. Disseminated dermal leishmaniasis caused by Leishmania siamensis in a systemic steroid therapy patient. Am J Trop Med Hyg. 2014 Nov;91(5):869-870. doi: 10.4269/ajtmh.13-0711.

2. Multiple fibrotic lesions in HIV cases:

Chiewchanvit et al. Chronic generalized fibrotic skin lesions from disseminated leishmaniasis caused by Leishmania martiniquensis in two patients from northern Thailand infected with HIV. Br J Dermatol. 2015 Sep;173(3):663-70. doi: 10.1111/bjd.13812.

3. Disseminated sporotrichoid lesions, and the first report of mucocutaneous involvement in HIV-infected patient:

Srivarasat et al. Case Report: Autochthonous Disseminated Cutaneous, Mucocutaneous, and Visceral Leishmaniasis Caused by Leishmania martiniquensis in a Patient with HIV/AIDS from Northern Thailand and Literature Review. Am J Trop Med Hyg. 2022 Nov 14:tpmd220108. doi: 10.4269/ajtmh.22-0108.

Additionally, another Mundinia member, L. orientalis, might result in a simple solitary lesion, possibly making clinician misdiagnose.  The reference for L. orientalis is here:

4. Jariyapan et al. Leishmania (Mundinia) orientalis n. sp. (Trypanosomatidae), a parasite from Thailand responsible for localised cutaneous leishmaniasis. Parasit Vectors. 2018 Jun 18;11(1):351. doi: 10.1186/s13071-018-2908-3.

5. Anugulruengkitt et al. Case Report: Simple Nodular Cutaneous Leishmaniasis Caused by Autochthonous Leishmania (Mundinia) orientalis in an 18-Month-Old Girl: The First Pediatric Case in Thailand and Literature Review. Am J Trop Med Hyg. 2022 Nov 21:tpmd220385. doi: 10.4269/ajtmh.22-0385.

After completing the review with more unusual clinical information of Mundinia species, this paper would be the most updated and greatly facilitate the clinical comprehension of this neglected disease.

All the best

Author Response

Response: Thank you for the suggestions. The modified version of the review has the updated information on the update classification of the Leishmania parasite. Various clinical reports have been added at an appropriate place in the manuscript.

Round 2

Reviewer 1 Report (Previous Reviewer 2)

I have read your manuscript and I appreciate the vast work you have done. With the changes introduced, I would recommend the publication of this study.

This manuscript is a resubmission of an earlier submission. The following is a list of the peer review reports and author responses from that submission.

Round 1

Reviewer 1 Report

The authors of the manuscript entitled "Unusual observations in leishmaniasis – an overview" present in a very didactic way the atypical observations of leishmaniasis, a parasitic disease considered a neglected tropical disease (NTD) by the Center for the Disease Control and Prevention (CDC).

A complete and detailed review of this parasitic disease is an excellent contribution to the scientific community.

The text had some minor errors that perhaps can be considered:

Lanes 184, 192, 197, 217. Replace immune-compromised with immunocompromised.

Lane 193. Replace socio-economic with socioeconomic.

Author Response

Thank you for the suggestions.

Response: As suggested, the corrections are incorporated at the respective places.

Reviewer 2 Report

Thank you for the opportunity to revise the manuscript sent by Drs Priya Yadav, Mudsser Azam, Venkatesh Ramesh, and Ruchi Singh about atypical leishmaniasis. Although the intention of the authors is laudable, I would like to point out some comments:

In my opinion, the length of the article should be adjusted. A very long introduction with detailed information duplicated in the text and tables makes the text hard to read.

The methodology used to do this comprehensive review is not explained. The authors have put lots of examples all together but there is no information on which characteristics have they used to consider an “atypical” case, and which parameters have been used to do the search. Which database have they consulted and which are the criteria to select one or another publication.

Moreover, I would recommend a better structure for this paper. The information given is vast. They have not discussed or tried to organize the unusual presentations considering risk factors or patient characteristics. Probably, there are common mechanisms to explain this rare presentation as immune impairment or new therapies but they have only collected the published information with no discussion.  It makes the reader feel like this is only a random list of atypical cases with no connection.

In summary, although the topic is of high interest, I would recommend the authors reconsider how they listed and structured the article to consider it for publication. 

Author Response

Comment1: The methodology used to do this comprehensive review is not explained. The authors have put lots of examples all together but there is no information on which characteristics have they used to consider an “atypical” case, and which parameters have been used to do the search. Which database have they consulted and which are the criteria to select one or another publication.

Response:

“Comprehensive literature search was performed on PubMed, Google and EuropePMC database with the search term “atypical leishmaniasis”, “unusual leishmaniasis”, “atypical visceral leishmaniasis”, “unusual visceral leishmaniasis”, “atypical post-kala azar dermal leishmaniasis”, “unusual post-kala azar dermal leishmaniasis”, “atypical cutaneous leishmaniasis”, “unusual cutaneous leishmaniasis”, “atypical mucocutaneous leishmaniasis” and “unusual mucocutaneous leishmaniasis”.

 The above statement is added to the manuscript now.

Comment 2: Moreover, I would recommend a better structure for this paper. The information given is vast. They have not discussed or tried to organize the unusual presentations considering risk factors or patient characteristics. Probably, there are common mechanisms to explain this rare presentation as immune impairment or new therapies but they have only collected the published information with no discussion. It makes the reader feel like this is only a random list of atypical cases with no connection. In summary, although the topic is of high interest, I would recommend the authors reconsider how they listed and structured the article to consider it for publication. 

Response: We agree that the information is vast; however, the presentation is not a random list of atypical cases. We have organized the review according to the disease forms, viz. VL, PKDL, CL and MCL. Further, each disease form was organized according to the respective unusual observations in the patients. A common mechanism is immune response impairment due to HIV or other coinfection, which is associated with all the forms and mentioned in the relevant places. It is challenging to apply the uniform review structure to all disease forms; hence, the disease forms are discussed separately; e.g. VL was organized based on immunocompromised, immunocompetent patients, patients with LD bodies in unusual body parts etc. Unusual PKDL cases were first grouped with disseminated or localized lesions and then the organs involved. CL cases were grouped as per unusual infective species, morphology, site (organs affected), number of lesions, etc. After describing disease forms, the last paragraph in each section summarizes the differential diagnosis for such cases.

Reviewer 3 Report

Overall the review  article by  Yadav et al is interesting, detailed and timely since it is becoming clear that leishmania genotypes and phenotypes are able to change and this is important for all aspects of this field including surveillance, diagnosis and treatment. There are few if any review articles covering the detailed pathology of atypical leishmaniasis and this article is a welcome addition to address this aspect of leishmaniasis. Overall the article is clearly written and well organized into the major disease sections including VL, CL, MCL and PKDL.

 The article starts off with and interesting perspective in the early history of leishmaniasis that is appropriate for this review. A number of typical and atypical disease presentations is presented in the text and tables highlighting different pathologies and epidemiology of Leishmania infection. This demonstrates the wide range of pathologies associated with this infection, and co-infections/other indications; information that will be useful to practitioners for diagnose and treatment these infections. 

Line 304, After treatment PBMC “start” producing IL-10. I believe this should be “stop” producing IL-10.

The authors should include more information about hybrid parasites. For example, the authors should however include the recent study by Lypaczewski in Lancet Microbe 2021  showing the ability of L. donovani to form hybrids with L. major and L. tropica and this has resulted in an epidemic if cutaneous leishmaniasis in Sri Lanka. The same authors have also identified L. donovani hybrids in Himachel India  (Lypaczewski iScience 2022) in a cutaneous leishmaniasis patient.

Author Response

Thank you for the suggestions.

Comment 1: Line 304, After treatment PBMC “start” producing IL-10. I believe this should be “stop” producing IL-10.

Response.

IL-10 production is stopped when VL patients are treated and recovered, where as high IL-10 levels in plasma and PBMC cultures are indicative of PKDL and in the said line, the pathogenesis of PKDL is described where after VL treatment PBMC’s start producing IL-10.

To clarify, the sentence is modified to

“After treatment of VL, peripheral blood mononuclear cells (PBMC) start producing IL-10, transforming from a T helper type 2 (Th2) to T helper type 1 (Th1) or a mixed Th2/Th1 immune response resulting in PKDL.”

Comment 2: The authors should include more information about hybrid parasites. For example, the authors should however include the recent study by Lypaczewski in Lancet Microbe 2021 showing the ability of L. donovani to form hybrids with L. major and L. tropica and this has resulted in an epidemic if cutaneous leishmaniasis in Sri Lanka. The same authors have also identified L. donovani hybrids in Himachal India  (Lypaczewski iScience 2022) in a cutaneous leishmaniasis patient.

Response

As suggested the information has now been added at relevant places

Added in Section 2

The interspecies and intraspecies genetic exchange has been documented experimentally and in nature and may contribute to new phenotypic traits. Apart from contributing to genome diversity of the parasite, interspecies hybrids of Leishmania such as those formed between L. donovani and L. majorL. tropica lead to the emergence of novel genotypes of Leishmania, consequently affecting the transmission of CL in countries like Sri Lanka(Lypaczewski and Matlashewski 2021 Ref 49).

Added in Section 3.3.1

  1. donovani strains to cause CL in Sri Lanka were found to contain gene polymorphism homologous to L. major and L. tropica genomes and were quite distinct from L. donovani causing VL in India. Further, intraspecies hybridization in L. donovani has been demonstrated to cause a phenotypic reposition from visceral to cutaneous form, causing an atypical CL phenotype. The recombinant derived from a CL focus in Himachal Pradesh resulted from genomic hybridization between two parental types of L. donovani, belonging to the Yeti ISC1 variant of Nepalese highlands (Lypaczewski et al Ref 141).

Round 2

Reviewer 2 Report

Thank you for your prompt response. Although there is a huge work on this work and some changes have been introduced to the manuscript, the general idea of having a vast list of cases with few discussion, in my opinion, makes it unsuitable for publication in this journal. There is a lot of repeated information between tables and text and its extension makes it hard to read.

I would consider maintaining tables with the summary of each case and within the text, only give some examples and summarize possible explanations of such presentations to make it more interesting for readers.

Author Response

As suggested, the tables are maintained with the summary of each case, and repetitive information has been condensed in the text.